# A Mini-Review: Biowaste-Derived Fuel Pellet by Hydrothermal Carbonization Followed by Pelletizing

Rhea Gallant [1], Aitazaz A. Farooque [1,2], Sophia He [3] , Kang Kang [4] and Yulin Hu [1,*]

1   Faculty of Sustainable Design Engineering, University of Prince Edward Island, Charlottetown, PE C1A 4P3, Canada
2   School of Climate Change and Adaption, University of Prince Edward Island, Charlottetown, PE C1A 4P3, Canada
3   Department of Engineering, Dalhousie University, Truro, NS B2N 5E3, Canada
4   Biorefinery Research Institute, Lakehead University, Thunder Bay, ON P7B 5E1, Canada
*   Correspondence: yulinhu@upei.ca

**Abstract:** This review article focuses on recent studies using hydrothermal carbonization (HTC) for producing hydrochar and its potential application as a solid fuel pellet. Due to the depletion of fossil fuels and increasing greenhouse gas (GHG) emissions, the need for carbon-neutral fuel sources has increased. Another environmental concern relates to the massive amount of industrial processing and municipal solid waste, which are often underutilized and end up in landfills to cause further environmental damage. HTC is an appealing approach to valorizing wet biomass into valuable bioproducts (e.g., hydrochar), with improved properties. In this review, the effects of the main HTC reaction parameters, including reaction temperature, residence time, and feedstock to water ratio on the properties and yield of hydrochar are described. Following this, the pelletizing of hydrochar to prepare fuel pellets is discussed by reviewing the influences of applied pressure, processing time, pellet aspect ratio, moisture content of the hydrochar, and the type and dosage of binder on the quality of the resulting fuel pellet. Overall, this review can provide research updates and useful insights regarding the preparation of biowaste-derived solid fuel pellets.

**Keywords:** hydrothermal carbonization; pelletizing; biomass; organic waste; hydrochar; fuel pellet

## 1. Introduction

At present, climate change, global warming, the depletion of fossil fuels, and the growing global population are some of the serious challenges faced worldwide. In this regard, many countries have launched stringent environmental regulations and plans to tackle these issues. For example, the Canadian government aims to reduce GHG emissions by 40–45% below 2005 levels by 2030 and to achieve net-zero GHG emissions by 2050, with multiple scenarios designed to reach these goals that involve phasing out fossil fuels and introducing many more renewable energy sources, such as wind, hydropower, and solar energy. Aside from these renewable energy sources, biomass is also an essential aspect of the future of Canadians [1]. Biomass is a renewable, sustainable, and abundantly available source that can be used for replacing fossil fuels and producing a range of value-added bioproducts such as bio-oil, biodiesel, bioethanol, and biogas [2]. So far, the investigated biomass conversion technologies can be broadly divided into chemical methods (e.g., transesterification), biological methods (e.g., anaerobic digestion), and thermochemical methods (e.g., pyrolysis, gasification, and combustion). By selecting the appropriate conversion method, low or even negative value of biomass and organic waste from industry and households can be valorized into value-added fuels, materials, and chemicals. This has been extensively reviewed by many researchers such as Singh et al. [3], Borrero-López et al. [4], Igbokwe et al. [5], Mahari et al. [6], Pattnaik et al. [7], and Shen and Sun [8].

Biomass is often regarded as an underutilized energy source. This is often due to some types of biomass such as microalgae, macroalgae, industrial processing waste, and municipal solid waste (MSW) containing a very higher water content, and thus it might not be energy-efficient to use conventional biomass conversion technologies such as pyrolysis, combustion, and gasification because energy-intensive drying is a necessary pre-treatment stage [9]. To solve this problem, researchers have been focusing on using hydrothermal treatments such as hydrothermal carbonization (HTC), hydrothermal liquefaction (HTL), and supercritical water gasification (SCWG), which mainly produce hydrochar, bio-oil, and $H_2$-rich syngas, respectively [10–12]. Among the investigated hydrothermal treatments, HTC is the main technology used to prepare hydrochar, a carbon-rich material, and can be used for different purposes. The resulting hydrochar can further undergo pelletizing to produce fuel pellets, which can be either used for domestic heating or for co-firing power plants. Aside from the fuel applications, hydrochar can also be activated by either chemical or physical methods to prepare an adsorbent to remove contaminants from wastewater such as heavy metals [13] and dyes [14] and for $CO_2$ capture [15]. To date, some previous literature has reviewed the use of hydrochar for improving plant growth [16], facilitating biomethane production in anaerobic digestion (AD) [17], heavy metal removal [18], gas cleanup [18], and as a soil amendment [19]. This present mini-review has focused on the energy application of hydrochar and is structured as follows: (i) an overview of the influences of the main reaction parameters and characteristics of biomass on the yield and properties of hydrochar; (ii) a description of the pelletizing of hydrochar to prepare fuel pellets; and (iii) a discussion of the most important properties for determining the fuel quality of the pellets derived from biomass.

## 2. HTC

### 2.1. Overview of HTC

HTC is a viable approach to converting wet biomass into a more valuable and energy-dense fuel source. HTC process occurs when biomass is subjected to the appropriate processing parameters such as temperature, residence time, feedstock loading, and pressure, in which hydrochar is produced as the main product, along with the formation of a water-soluble phase and trace amounts of gaseous products [20]. Recent studies from 2020 to 2022 are summarized in Table 1.

**Table 1.** The summary of recent studies on HTC of biomass for hydrochar production.

| Feedstock | Reactor Configuration | Temp (°C) | Initial Pressure (MPa) | Residence Time (h) | Solid Loading (wt.%) | Main Conclusions | References |
|---|---|---|---|---|---|---|---|
| Glucose | Batch | 180–270 | / | 0–8 | 25 | • Glucose was completely decomposed at 180 <br> • At 220 °C, longer residence times led to the formation of nano/microsphere structures. <br> • The highest hydrochar yield (50.7 wt.%) was produced at 220 °C for 6 h. | [21] |
| Cellulose; xylan, dealkaline lignin, soybean protein isolate | Batch | 220 | 1 ± 0.1 MPa | 2 | 9.1 | • The synergistic effect of different biomass components on hydrochar was reduced when the blending ratio of lignocellulose components increased. <br> • The formation of nitrogenates in the liquid phase was dominant due to the Maillard and Mannich reactions. <br> • Maximum hydrochar yield (54.2 wt.%) was obtained through co-HTC of protein and cellulose. | [22] |
| Corn straw | Batch | 220 | 1.2 MPa | 0.5–2 | 16.7 | • The liquid phase from HTC was reused as the water source to prepare feedstock slurry for HTC. <br> • The presence of phenol, furfural, and 5-HMF in the liquid phase promoted the polymerization reaction and led to an increase in the hydrochar yield and HHV. <br> • The presence of acetic acid and 2,5-hexanedione in the liquid phase caused a decrease in the hydrochar yield. | [23] |
| Mixture of sewage sludge (S) and pinewood sawdust (L) | Batch | 160–280 | / | 0–4 | 5.88–20 | • The liquid phase was used in anaerobic digestion to produce $CH_4$. <br> • Increasing the percentage of sawdust in the feedstock mix caused an increase in the hydrochar yield and its energy content. <br> • Increasing the temperature lowered the hydrochar yield but increased its HHV. | [24] |

**Table 1.** *Cont.*

| Feedstock | Reactor Configuration | Temp (°C) | Initial Pressure (MPa) | Residence Time (h) | Solid Loading (wt.%) | Main Conclusions | References |
|---|---|---|---|---|---|---|---|
| Waste-activated sludge | Batch | 170–230 | / | 5–60 min | / | • Adding 0.1–0.5 M citric acid/HCl to the HTC process helped the solubilization of N and P in the liquid phase.<br>• At 230 °C, 15 min, and 0.5 M HCl, it was found that ~100% of the N and ~94% of the P in the water phase existed in the form of $NH_4$ and $PO_4$, respectively.<br>• The highest yield of hydrochar (56.2 wt.%) was produced at 170 °C. | [25] |
| Wheat straw | Batch | 160–240 | / | 1 | 9.09 | • Pre-washing removed 90% of the K and P from wheat straw before HTC, thereby reducing the risk of slagging and fouling.<br>• Hydrochar obtained at 200 °C had a similar ignition temperature to bituminous coal. | [26] |
| Corn straw | Batch | 180–240 | / | 30–90 | 13.04 | • C was primarily retained in the solid phase.<br>• Increasing the temperature lowered the O content in the hydrochar, along with a decrease in the O/C and H/C atomic ratios.<br>• Compared with temperature and residence time, particle size had a minor effect on hydrochar production.<br>• The maximum hydrochar yield of 67.6 wt.% was achieved at 180 °C. | [27] |
| Corncobs and peanut residue | Batch | 180–260 | 0.3 | 1–4 | 9.09 | • Peanut residue was a more suitable feedstock for fertilizer preparation than corncobs.<br>• Co-HTC improved the N recovery rate from 8.52% to 19.51%.<br>• At higher temperatures, N tended to migrate to the hydrochar and the oil phase.<br>• P was fixed in the hydrochar, and the fixation of P and N in the hydrochar could be improved by modifying the pH value.<br>• The highest hydrochar yield obtained from corncobs and peanut residue was 51.2 wt.% and 22.5 wt.%, respectively. | [28] |

**Table 1.** *Cont.*

| Feedstock | Reactor Configuration | Temp (°C) | Initial Pressure (MPa) | Residence Time (h) | Solid Loading (wt.%) | Main Conclusions | References |
|---|---|---|---|---|---|---|---|
| Rice husk | Batch | 220 | 0.5 | 1 | 20 | • The liquid phase was reused as the reaction medium five times.<br>• The recycled liquid phase promoted the formation of oxygen-containing functional groups of hydrochar.<br>• The recycled liquid phase promoted dehydration and decarboxylation, resulting in a decrease in the H/C and O/C atomic ratio of hydrochar.<br>• The recycled liquid phase enhanced the adsorption capacity of hydrochar for malachite green.<br>• A maximum hydrochar yield of 76.6 wt.% was obtained at the third time of recycling. | [29] |
| Bamboo | Batch | 200 | / | 24 | 16.7 | • The hydrochar modified by dithiocarbamate was effective to adsorb Pb(II).<br>• The surface complexation was found to be the main mechanism for Pb(II) adsorption. | [30] |

The hydrochar obtained through HTC of biomass is a carbon-rich material and has been shown to have improved properties compared with the original biomass, e.g., energy density, grindability, and improved overall physicochemical characteristics. Hydrochar has been shown to have an increased hydrophobicity, thereby easing the stages of storage and transportation [31]. Typically, the high moisture content of the fuel sources results in a lower calorific value and creates challenges for storage and transportation. In addition, the chemical structure and energy content of hydrochar are similar to those of natural coal, making it suitable for using as a solid fuel in conventional combustion processes. During the HTC process, a series of reactions such as hydrolysis, depolymerization, decarboxylation, and condensation can occur, and the underlying reaction mechanisms and kinetics of HTC have been investigated by Yang et al. [32] and Ischia et al. [21]. The properties of hydrochar are highly dependent on the characteristics of the biomass and the processing parameters, including temperature, residence time, and solid loading (also known as the feedstock to water ratio), as discussed in the following sections.

### 2.2. Characteristics of the Biomass

Some difficulties facing the introduction of biomass as a solid fuel source are caused by the complexity of the biochemical composition of the biomass. Generally speaking, biomass primarily contains cellulose, hemicellulose, and lignin, and lipids and proteins can also be observed, particularly for microalgae and macroalgae. Table 2 summarizes the biochemical composition of some common types of biomass. Each component has an influence on the properties of the biomass as a fuel source. The degradation reaction pathway of glucose (a monomer of the cellulose) is shown in Figure 1. Other variables to consider in biomass include the ash content, which primarily consists of inorganics. The inorganic fraction might demonstrate either a positive or a negative catalytic impact on the HTC of biomass and hydrochar production. For example, some inorganics present in the original biomass might demonstrate positive catalytic effects. However, ash fouling problems might occur and lead to blockage of the reactor and a lower mass and heat transfer.

**Table 2.** Summary of the biochemical composition of some types of biomass.

| Biomass Type | Lignin | Cellulose | Hemicellulose | Extractives | Reference |
|:---:|:---:|:---:|:---:|:---:|:---:|
| Empty fruit bunches | 23.87% | N/A | N/A | 4.29% | [33] |
| Food waste | 5.78% | N/A | N/A | N/A | [34] |
| Food waste | 28.80% | 45.30% | 3.30% | N/A | [35] |
| Yard waste | 19.10% | 38.80% | 25.20% | N/A | [35] |
| Yard + food waste | 23.95% | 42.05 | 14.25% | N/A | [35] |
| Ginko leaf residues (GLR) | 17.90% | 29.10% | 45.20% | N/A | [36] |
| Oat husk | 22.60% | 19.36% | 50.51% | 7.53% | [37] |
| Pine sawdust | 30.00% | 42.21% | 25.00% | 2.79% | [37] |

As mentioned earlier, HTC process involves a series of reactions, including hydrolysis, dehydration, decarboxylation, condensation, polymerization, and aromatization, and these reactions do not occur consecutively but in parallel. It is undoubtedly the nature of these reactions and their significances in the HTC process varies among different types of biomass [38]. Wilk et al. [39] conducted a comparative study on the HTC of two types of lignocellulosic biomass (i.e., pine and acacia, a coniferous and a deciduous wood) and sewage sludge at 200 °C for 4 hours with a 1:8 biomass to water weight ratio. An increase in the fixed carbon content was observed in the hydrochar when woody biomass was used as the feedstock, while carbon tended to be transferred to the water phase rather than the solid phase in the HTC of sewage sludge. In a thermogravimetric analysis of hydrochar, it was found that the activation energy of sewage-sludge-derived hydrochar was 50% lower than that of woody-biomass-derived hydrochar. Kabakci and Baran [40] applied six

different types of biomass including wood sawdust, olive pomace, walnut shells, apricot seeds, tea stalks, and hazelnut husks in a HTC reactor at 220 °C for 90 min with a 1:4 biomass: water weight ratio. The results showed that the hydrochar obtained from HTC of olive pomace had the highest carbon content and heating value, and a low ash content. In terms of the ignition temperature, woody biomass and its derived hydrochar exhibited the highest ignition temperature, even though both had the highest amount of volatile matter. This could be owing to the differences in the ignition temperature of the volatiles released from the wood sawdust [41]. The main differences between original feedstock and the hydrochar in terms of their compositions (volatile matter, fixed carbon, and ash content) are summarized in Table 3. Aside from the characteristics of biomass, the temperature, residence time, and feedstock to water ratio are other critical reaction parameters and are discussed below.

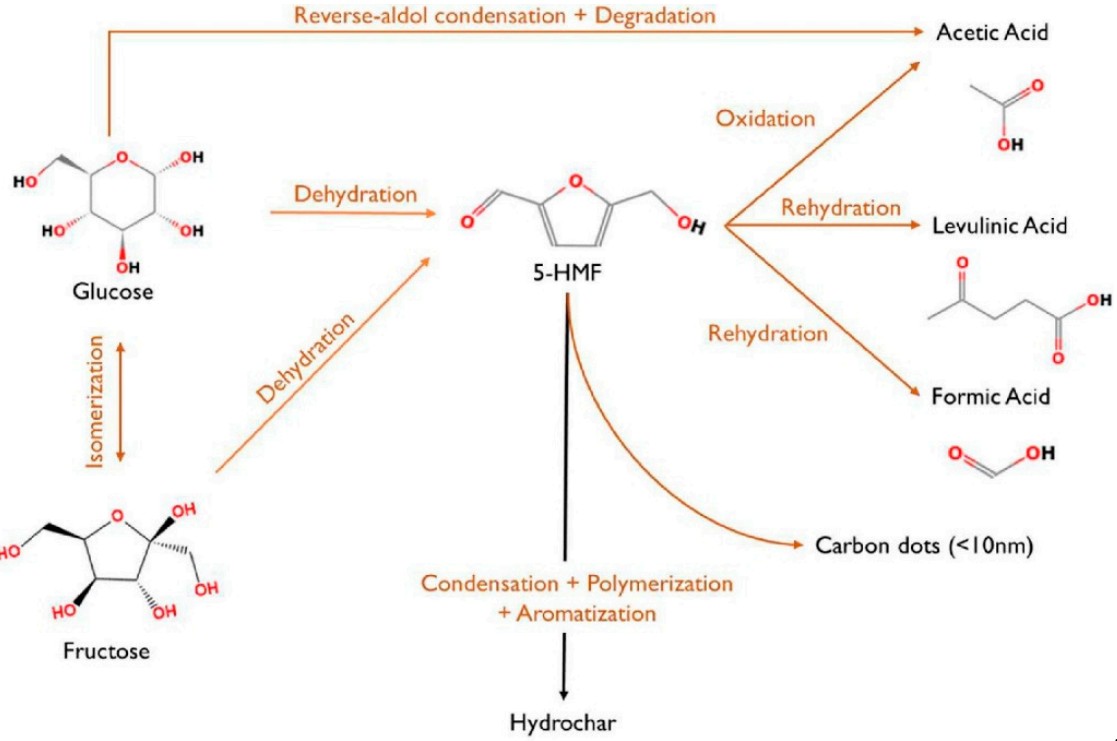

**Figure 1.** The possible reaction pathway of glucose during HTC [21] (with copyright permission from Elsevier).

**Table 3.** Comparison between the feedstock and the resulting hydrochar in terms of volatile matter (VM), fixed carbon (FC), and ash content.

| Feedstock | Before HTC | | | After HTC | | | Reference |
|---|---|---|---|---|---|---|---|
| | VM (%) | FC (%) | Ash (%) | VM (%) | FC (%) | Ash (%) | |
| Sewage sludge | 44.01 | 1.17 | 54.82 | 51.72 | 27.84 | 20.44 | [24] * |
| Pinewood sawdust | 82.07 | 16.65 | 1.28 | | | | |
| Corncobs | 79.69 | 16.26 | 4.05 | 97.5–99.74 | / | 0.26–2.5 | [28] |
| Peanut residue | 80.65 | 12.56 | 6.84 | 88.47–96.01 | / | 3.99–11.53 | [28] |
| Rice husk | 51.04 | 30.61 | 18.35 | 39.48–42.70 | 36.36–39.19 | 20.94–21.55 | [29] |
| Food waste | 72.55 | 15.98 | 11.47 | 49.62–59.96 | 28.85–40.15 | 8.45–11.35 | [34] |

* A mixture of sewage sludge and pinewood sawdust was used as the feedstock during HTC at 250 °C.

### 2.3. Temperature

Temperature is the most significant parameter in the HTC process, and it plays an important role in determining the energy demand. The carbonization reaction favors temperatures up to 250 °C. When the temperature increases further up to 375 °C, the formation of bio-oil is dominant, while gas formation becomes dominant at temperatures above 375 °C. This variation is mainly related to the properties of water in different temperature regions, as illustrated in Figure 2.

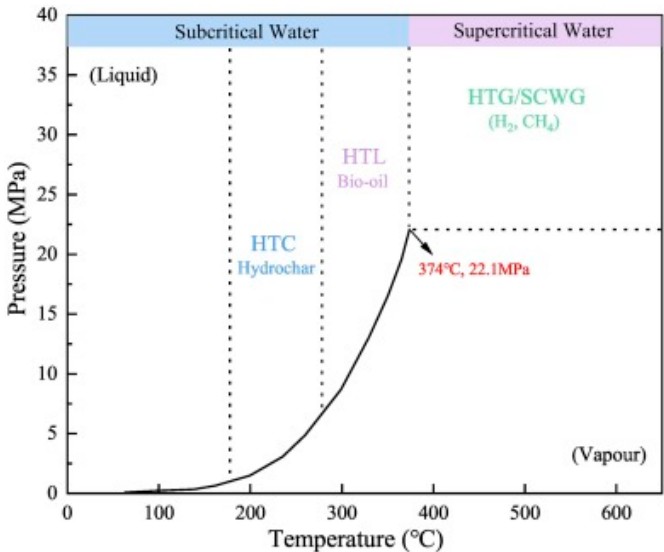

**Figure 2.** Effects of temperature and pressure on the properties of water [42] (with copyright permission from Elsevier).

At subcritical conditions, the concentrations of $H_3O^+$ and $OH^-$ are higher than those of water under normal conditions, and hence the acidic and basic reactions can be promoted. In addition, a lower dielectric constant of water is expected to be achieved at subcritical conditions compared with that of water at ambient conditions, making water behave in a similar way to an organic solvent, and thus some water-insoluble chemicals such as fatty acids can be dissolved [43].

In terms of the HTC process, the hydrolysis of biomass molecules is predominant, and the breakage of their chemical bonds can lead to the formation of small molecules and intermediates. If the temperature increases further, the polymerization and aromatization reactions between small molecules are facilitated. As a result, the selection of the temperature of HTC greatly affects the yield and properties of the hydrochar. Wiedner et al. [44] investigated the effect of temperature on the hydrochar yield obtained through HTC of poplar wood, olive residues, and wheat straw at 180–230 °C for 8 h with a feedstock:water weight ratio of 3:10 (Figure 3). Increasing the temperature of HTC decreased the yield of hydrochar, since the more volatile fraction of biomass was decomposed to form water-soluble compounds and then was partitioned into aqueous phase. This trend is consistent with our previously published study, where spent coffee grounds (SCG) were carbonized in an HTC reactor at 150–210 °C for 30 min and the feedstock to water weight ratio was 1:5. The influence of temperature on co-HTC of *Miscanthus* and coal for hydrochar production was evaluated at 200–260 °C by Saba et al. [45]. These authors found that the yield of hydrochar obtained from co-HTC decreased with an increase in temperature. The decrease in the hydrochar yield could be due to depolymerization of the biomass major components, especially for hemicellulose and cellulose, since the depolymerization temperature of lignin is significantly higher than that of hemicellulose and cellulose. The thermal degradation of cellulose and hemicellulose could lead to the formation of water-soluble organic acids and thus would lower the yield of hydrochar but increase the yield of the aqueous phase [46].

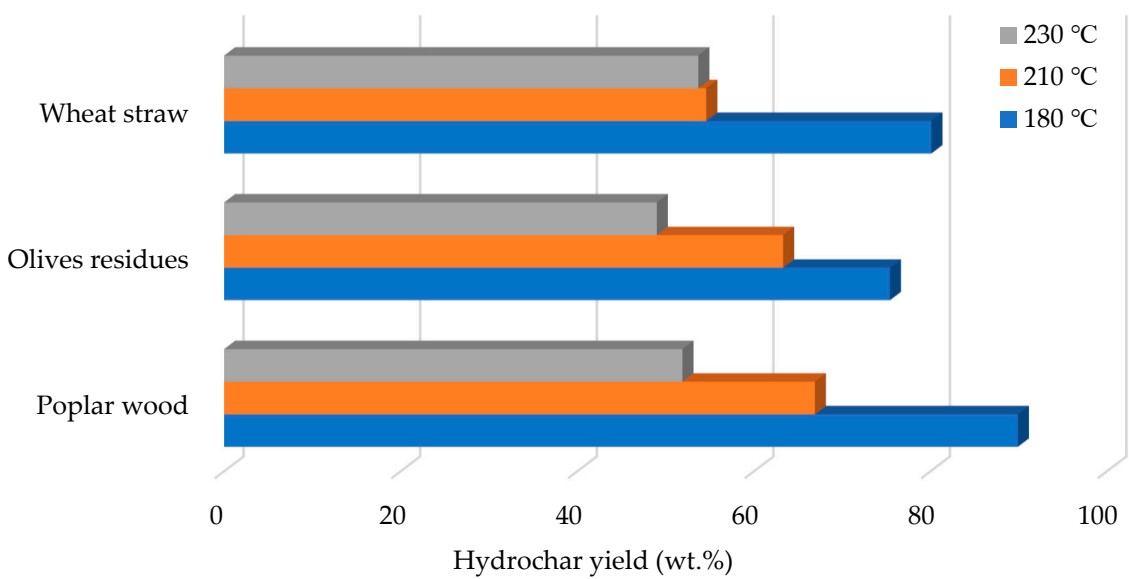

**Figure 3.** Effect of temperature on the yield of hydrochar [44].

The temperature of HTC also significantly influences the properties of hydrochar. Nakason et al. [47] found that the ash content of hydrochar increased with an increase in the temperature, which can be explained by: (i) other biomass constituents such as volatiles being given off as gases at higher temperatures, which might lead to a higher proportion of ash in the hydrochar; and (ii) some inorganics might be re-adsorbed on the surface of the hydrochar, since hydrochar can exhibit a porous structure at higher HTC temperatures. On the other hand, it should be noted that the ash or inorganic fraction of the original biomass could be partitioned into the water phase during the HTC treatment. Moreover, the C content of hydrochar increased with an increase in the temperature, along with a decrease in the O content, and this resulted in a higher heating value at higher temperatures. As well as temperature, residence time is another very important parameter affecting the yield and properties of hydrochar, as described below in Section 2.3.

*2.4. Residence Time*

The residence time is defined as the time for which the raw material is held at the designated temperature. This factor affects not only the products' distribution and properties but also the energy balance and operating costs. Typically, the carbonization reaction is promoted by a prolonged residence time. As indicated in the literature, the investigated residence times vary from few minutes to hours, and the most commonly studied residence time is 60 min [48]. Khoo et al. [49] found that the yield of hydrochar was inversely proportional to the residence time, which is in agreement with previous studies conducted by Wang et al. [50] and Shrestha et al. [51]. A more dramatic reduction in the yield of hydrochar can be observed with extended residence times, particularly when the temperature is above 200 °C. This is because of the increased decomposition of large molecular weight compounds to water-soluble compounds rather than the retention of these in the solid phase [49]. Regarding the influence of the residence time on the properties of hydrochar such as HHV, it was observed that the HHV of hydrochar increased with a longer residence times and lower temperatures (such as 180 °C) due to an increase in the C content; however, a drop in the HHV of hydrochar was found with prolonged residence times and higher temperatures [52]. The influence of residence time on the surface morphology of hydrochar was studied by Shao et al. [53], and the results indicated that an increase in the residence time was favorable for developing a porous structure, as depicted in Figure 4, which might give the hydrochar a porous structure so it could be used as an adsorbent.

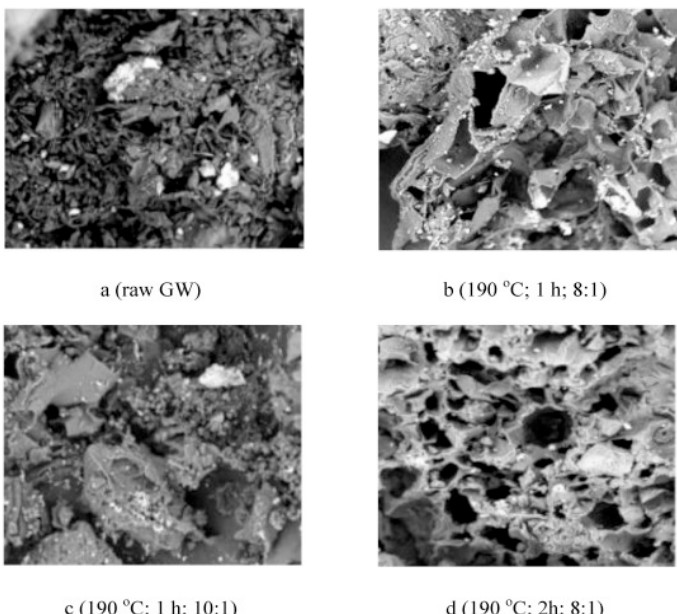

a (raw GW)  b (190 °C; 1 h; 8:1)

c (190 °C; 1 h; 10:1)  d (190 °C; 2h; 8:1)

**Figure 4.** SEM images of green waste and the resulting hydrochar for different residence times [53] (with copyright permission from Elsevier).

Clearly, there is a relationship between the residence time and the temperature, and thus, it is more useful to evaluate the effect of the severity factor on HTC instead of studying the individual effects of temperature and residence time on HTC. The severity factor is defined as the impact of residence time and temperature on the HTC of biomass and can be calculated using the following equations (Equations (1) and (2)) [51,54,55].

$$Ro = t \times e^{\left(\frac{T-100}{14.75}\right)} \tag{1}$$

$$Severity\ factor = \log Ro \tag{2}$$

where *Ro* is the reaction ordinate, *t* is the residence time (min), and *T* is the temperature (°C).

By calculating the severity factor, Hoekman et al. [55] reported that no energy densification occurred during the HTC of loblolly pine when the severity factor was lower than 4, but a sharp increase in the energy content of hydrochar was found at a severity factor of 5–6, which was accompanied by an additional increase in the energy content of the hydrochar when the severity factor was above 6. In another study, severity factors ranging from 4.3 to 5.3 were found to result in the greatest increase in the energy content of hydrochar during the HTC of sawdust, and the effect of the severity factor on the HHV of hydrochar was minor when the severity factor was above 5.8 [56]. In terms of the influence of the severity factor on the yield of hydrochar, Jeder et al. [57] found that the yield of hydrochar dropped continuously with an increase in the severity factor from 4.0 to 7.6. This decrease in the yield of hydrochar was accompanied by carbon enrichment and a reductions in the O and H contents.

### 2.5. Feedstock: Water Ratio

The feedstock: water ratio is another very critical parameter of the HTC process. Lower water content during HTC could lead to and uneven temperature distribution inside the reactor, while a higher water content during HTC promotes biomass degradation, similar to the hydrolysis reaction. Moreover, feedstock loading and water content play a role in determining the quality of the hydrochar [48]. The effects of temperature (*T*), time (*t*), and the feedstock to water ratio (F:W) on the proximate analysis (fixed carbon, FC; volatile matter, VM; and ash), ultimate analysis, and HHV of hydrochar are summarized in Table 4.

**Table 4.** Effects of temperature (*T*), feedstock to water weight ratio (F:W), and residence time (*t*) on the properties of hydrochar.

| Feedstock | *T*(°C) | F:W Weight Ratio | *t*(h) | Proximate Analysis (%) | | | Ultimate Analysis (%) | | | | HHV (MJ/kg) | Reference |
|---|---|---|---|---|---|---|---|---|---|---|---|---|
| | | | | FC | Ash | VM | C | H | O | N | | |
| Food waste | 200–260 | 1:8 | 1–4 | 28.85–40.15 | 8.61–11.35 | 49.62–59.8 | 66.11–72.97 | 6.83–7.11 | 5.53–12.98 | 2.41–3.87 | 28.88–32.36 | [34] |
| Food waste/molasses | 200–260 | 1:8 | 1–4 | 24.78–31.85 | 10.57–13.47 | 54.68–62.83 | 62.1–68.26 | 6.92–7.34 | 7.51–15.87 | 1.8–3.84 | 27.31–30.46 | [34] |
| Food waste/lime + molasses | 200–260 | 1:8 | 1–4 | 19.79–31.02 | 17.55–19.66 | 51.56–60.89 | 57.6–63.49 | 6.11–6.72 | 7.98–13.58 | 2.56–3.35 | 25.22–27.68 | [34] |
| Food waste | 220 | 1:20 | 1 | 32.68 | 41.4 | 62.87 | 59 | 5.43 | 32.68 | 1.56 | 21.64 | [35] |
| Yard waste | 220 | 1:20 | 1 | 40.64 | 43.4 | 55.54 | 65.2 | 5.83 | 28.05 | 0.88 | 24.37 | [35] |
| Yard and food waste | 220 | 1:20 | 1 | 37.56 | 4.32 | 57.4 | 68.26 | 6.02 | 24.4 | 0.01 | 27.64 | [35] |
| Ginko leaf residues | 100–220 | 1:10 | 0.5 | 24–29 | 7.1–10.0 | 61.0–68.6 | 45.8–50.1 | 5.7–5.9 | 42.8–46.3 | 1.2–2.0 | 19.1–22.1 | [36] |
| Oat husk/pine sawdust | 175–235.4 | 1:12 | 0.5 | 28.31 | 0.74 | 70.95 | 53.29 | 5.9 | 39.68 | 0.35 | 19.18–21.5 | [37] |
| Rapeseed meal/pine sawdust | 175–235 | 1:8 | 0.5 | N/A | 1.35–1.53 | N/A | 51.88–53.13 | 6.21–6.22 | 37.05–38.93 | 1.83–2.25 | 20.87–22.07 | [58] |
| Municipal yard waste | 160–200 | 1:20 | 2–24 | 12–28% | 6–7 | 58–80 | 48–60 | 8–9 | 32–49 | 1–2 | 18.23–25.54 | [59] |

Hu et al. [10] evaluated the effect of the feedstock to water weight ratio (1:5 to 1:15) on the HTC of spent coffee grounds at 150 °C for 30 min, and the results indicated that the yield of hydrochar slightly reduced when more water was added. This trend might be due to the increase in biomass decomposition at a higher water content. It was also found that the effect of feedstock loading on HHV of hydrochar was insignificant. Nizamuddin et al. [60] also reported that a higher yield of hydrochar was obtained at a lower water during hydrothermally carbonization of palm shells at 200 °C for 5 min. After HTC treatment, biomass can be converted into energy-dense hydrochar, and it can further undergo pelletizing to prepare fuel pellets. The fuel pellets can be either used for domestic heating or in coal power plant. The main parameters affecting the pelletizing process and the key properties of the prepared fuel pellets are discussed below in Section 3.

*2.6. Catalyst*

A catalyst is often applied to facilitate the thermal degradation of biomass. Both homogenous and heterogenous catalysts have been used. Hydrolysis is a very important reaction involved in HTC and adding acid catalysts can promote hydrolysis. Aside from acid catalysts, alkaline catalysts such as carbonate, hydroxide, and bicarbonate have demonstrated beneficial impacts on bio-oil production by deterring the formation of char. Thus, alkaline catalysts have been commonly applied in bio-oil production processes such as HTL. Nevertheless, only few studies have applied catalysts in the HTC process for enhancing the production of hydrochar, including Evcil et al. [61], who used a mixture of $AlCl_3$ and HCl, and Wikberg et al. [62], who used $H_2SO_4$.

## 3. Pelletizing of Hydrochar

In the pelletizing process, the energy density of the hydrochar can be increased, and there are different parameters affecting the pelletizing process and the properties of the fuel pellets, such as applied pressure, the processing time, the pellets' aspect ratio, the moisture content of the hydrochar, and the type and dosage of the binder. These parameters should be optimized to ensure that the quality of the fuel pellets can meet the standards, such as moisture and ash content, and pellet size. Recent studies on pelletizing hydrochar to prepare fuel pellets are summarized in Table 5.

**Table 5.** Summary of the pelletizing parameters used for preparing hydrochar pellets.

| Feedstock | Pellet Diameter (mm) | Pellet Length (mm) | T (°C) | Applied Pressure (MPa) | Holding Time (s) | Moisture (%) | Binder | Reference |
|---|---|---|---|---|---|---|---|---|
| Food waste | 9.5 | 28 | 115 | 10 | 30 | 1.96–5.14 | Molasses/CaO | [34] |
| Mixture of food waste and yard waste | 10 | 90 | 90 | 250 | 30 | / | / | [35] |
| Municipal yard waste | 10 | 90 | 100 | 250 | 30 | 60–70 | / | [58] |
| Cornstalk | 8 | / | / | 13.6 | 120 | / | Lignin | [63] |
| Wheat straw | / | / | / | 130 | 30 | 13 | / | [64] |
| Tobacco stems | 5 | 39.5 | 80 | 20 | 30 | / | $K_2CO_3/CaCO_3$ | [65] |
| Fecal sludge | 10 | 30 | 105 | 12 | / | / | Lignin/starch/ $Ca(OH)_2$ | [66] |
| Anaerobic granular sludge | 10 | 90 | / | 150 | 30 | / | / | [67] |
| Mixture of food waste and coal | 13 | / | 105 | 16 | 30 | / | Molasses | [68] |

*3.1. Pelletizing Parameters*

(1) The moisture content of hydrochar: The moisture content of hydrochar affects the mechanical strength of the fuel pellet. A previous study concluded that a moisture content of 8–10% of hydrochar was ideal prior to pelletizing. However, it was found that a moisture content of 12–20% may help the pelletizing/densification process at room temperature. The

pelletizing/densification process was not able to accept a moisture content in the feed above 20%, as the pellet mill might become clogged. However, the authors also explained that a larger die diameter (e.g., 7.8 mm in diameter) was capable of pelletizing high-moisture feedstock (>10%) without clogging the pellet mill [66].

(2) Binder type and dosage: Previous studies have used different types of binder to prepare fuel pellets from hydrochar. Fakkaew et al. [63] investigated the effects of the type (i.e., lignin, starch, and $Ca(OH)_2$.) and dosage of the binder on the properties of fuel pellets prepared by HTC of fecal sludge. In general, starch was observed to be the best binder for making high-quality hydrochar pellets. The results showed that increasing the binder dosage lowered the energy content of the pellets, except when lignin and starch was used as the binder at a dosage of <10 wt.%. Adding a binder did not significantly affect the pellets' density, and the use of starch led to the highest density of the fuel (1050–1100 $kg/m^3$). Regarding the compressive strength of the pellets, a decrease in the strength of the pellets was observed from 8 to 6kN/$m^2$ when the lignin content was increased from 0 to 15 wt.%. On the contrary, an increase in the compressive strength of the pellets was found when the dosage of $Ca(OH)_2$ increased. Sharma et al. [67] used molasses as the binder and the dosage ranged from 5 wt.% to 30 wt.%, and the density of the pellets improved with the addition of molasses. This increase in pellet density was caused by the formation of a solid bridge by the addition of molasses, and the particles were tightly bonded and thus the density increased. Adding 30 wt.% molasses resulted in the highest density of 1683.24 $kg/m^3$ and highest energy density of 37.54 $GJ/m^3$. The mechanism of hydrochar pelletization using molasses as the binder was described by the authors, and it was observed that an increased contact surface area, the presence of oxygen-containing functional groups, and high compression pressure were the main reasons for the pelletization of hydrochar. Other previous studies have prepared fuel pellets without using any binder, such as Sharma and Dubey [58].

(3) Temperature, holding time, and applied pressure: A typical large-scale pellet mill is equipped with a heating element, and thus the feed can be partially dried before pelletization. Some studies applied a band heater to provide the heat if the pellet mill used did not have a heating element installed. Yan et al. [69] applied a temperature of 140 °C for heating the hydrochar, and the pressure and holding time were 160 MPa and 30 s, respectively. In the literature, lab-scale hydraulic pressers have been widely used, and some lab-scale hydraulic pressers are not equipped with a heating element, and thus the pressing process is conducted at room temperature, followed by curing the pellet in an oven for a certain period of time. Fakkaew et al. [66] initially applied 12 MPa using a hydraulic presser to a mixture of hydrochar and a binder, followed by drying at 105 °C for 12 hours to prepare pellets. In another study, a mixture of hydrochar, a binder (i.e., $K_2CO_3$ and $CaCO_3$), and water was compressed at 20 MPa and held at this pressure for 30 s. After compression, the sample was dried in an oven at 80 °C for 24 h to prepare pellets. A previous comprehensive review has summarized the effects of the pelletizing temperature on the properties of pellets [70]. Most previous studies have applied a holding time of 30 s in the pelletizing process; however, there are large differences in the applied pressure. Sharma and Dubey [58] used a very high applied pressure of 250 MPa to make pellets through HTC of municipal yard waste, since no binder was added. In other studies, much lower applied pressures have been utilized; for example, an applied pressure of 13.6 MPa was used by Zhu et al. [63] to prepare hydrochar pellets from cornstalks, and 130 MPa was applied to make fuel pellets from harvested wheat straw by Zhang et al. [64].

### 3.2. Properties of the Fuel Pellets

In Canada, standard fuel pellets are made from woody biomass and finished wood products. Some harvest residues have been utilized to a small extent [71]. Standards for solid biofuels have been developed by the International Organization of Standardization (ISO) within the 17,225 series and have been adopted by the Canadian Standards Association Group (CSA). These standards lay out the properties affecting the performance of

solid biofuels, including the size, shape, density, energy content, moisture content, and ash or non-combustible content [72]. For fuel pellets, the ash content must be in the range of 0.7–2%, depending on the grade of the wood pellet. Additional requirements state that the durability must be a minimum of 96.5%, on the basis of the weight. Although the standards vary slightly for industrial applications, the solid fuels derived from biomass should meet the same standards in order to be implemented for residential heating or industrial-scale applications. Currently, CSA ISO 17225-6 "Solid Biofuels-Fuel specifications and classes—Part 6: Graded non-woody pellets" describe the requirements for fuel pellets prepared from non-wood-based biomasses, including algae and food waste (https://www.iso.org/standard/76093.html; accessed on 1 September 2022); however, these standards currently do not include solid biofuels that are processed by HTC. The most critical parameters for wood pellets based on the CAN/CSA-ISO 17225 Part 2 Standard are presented in Table 6. Overall, a summary of the fuel pellets prepared through HTC of biomass is shown in Table 7.

**Table 6.** Summary of the key specifications of different grades of wood pellets (suitable for residential and commercial applications) based on the standard [71].

| Key Specifications | Unit | Grade A1 | Grade A2 | Grade B |
|---|---|---|---|---|
| Diameter | mm | $6 \pm 1$ or $8 \pm 1$ | $6 \pm 1$ or $8 \pm 1$ | $6 \pm 1$ or $8 \pm 1$ |
| Length | mm | 3.15–40 | 3.15–40 | 3.15–40 |
| Moisture | % of weight | $\leq 10$ | $\leq 10$ | $\leq 10$ |
| Ash | % of weight | $\leq 0.7$ | $\leq 1.2$ | $\leq 2.0$ |
| Durability | % of weight | $\geq 97.5$ | $\geq 97.5$ | $\geq 96.5$ |
| Fines content | % of weight | $\leq 1$ | $\leq 1$ | $\leq 1$ |
| High calorific value | MJ/kg | $\geq 18.6$ | $\geq 18.6$ | $\geq 18.6$ |
| Bulk density | kg/m$^3$ | 600–750 | 600–750 | 600–750 |

**Table 7.** Summary of the properties of hydrochar obtained through HTC of biomass.

| Feedstock | Density (kg/m$^3$) | Moisture Uptake (%) | Durability (%) | Reference |
|---|---|---|---|---|
| Yard waste | 1621 | 2.5 | / | [35] |
| Yard + food waste | 1678 | 7.5 | / | [35] |
| Oat husk/pine sawdust | / | 7.45 | 99.20 | [37] |
| Food waste | 872.1–936.8 | 2–5 | 98.6–99.0 | [34] |
| Food waste/molasses | 912.9–1095.6 | 3.7–9 | 98.6–99.9 | [34] |
| Food waste/lime + molasses | 920.1–1109.0 | 5.39–9.4 | 96.9–99.0 | [34] |
| Food waste | 1444.86 | 11 | / | [35] |
| Municipal ward waste | 1472.70–1661.59 | 1.67–55.18 | / | [58] |
| Empty fruit bunches | 970–1110 | 3.70–5.86 | / | [73] |
| Food waste/woody biomass | 1137–1365 | 1.99–6.52 | / | [74] |

(1) Bulk density: The bulk density of fuel pellets can be tested to provide guidance for sizing the storage space, along with a consideration of the energy consumption. An estimation of the bulk density can be made by weighing a known volume of the pellets, and void spaces between the fuel pellets must be avoided by shaking or tapping. Normally, the bulk density of fuel pellets should be higher than 600 kg/m$^3$ [71].

(2) Moisture resistance capacity: In the literature, the moisture resistance capacity of fuel pellets is typically determined by using a humidity chamber over a period of several hours. In comparisons of raw feedstocks with the hydrochar, there is often an increase in the hydrophobicity after the HTC treatment, which might be caused by the decomposition of hemicellulose. Hemicellulose is reported to have the highest capacity of water uptake compared with other lignocellulosic components such as cellulose and lignin. This might result from the large differences in the chemical structures of hemicellulose,

cellulose, and lignin [75]. Another reason for the increased hydrophobicity is related to the loss of hydroxyl groups, which act as a water adsorption agent in the biomass during hydrothermal treatment. Commonly, moisture resistance test results have indicated that the pretreated fuel pellets made from biomass have a higher resistance to moisture and are highly hydrophobic in nature compared with the original biomass. Correspondingly, such pellets can be safely stored for a long time without the risk of biological deterioration, and the transportation costs can be reduced [73].

(3) Pellet strength and durability: To meet the standards regarding the strength and durability of fuel pellets, multiple approaches have been used. However, so far, no standardized method has been developed for fuel pellets other than wood pellets. Some studies have used a tumbling can tester to characterize the pellets' durability, e.g., Zhai et al. [34], Sharma et al. [35], and Zaini et al. [73]. In a previous study, a procedure for determining the mechanical strength of coke was used to evaluate the hydrochar pellets produced through HTC of empty fruit bunches [73]. Briefly, the hydrochar pellets were placed inside a rotating drum, and then the drum started rotating. When the tumbling finished, the pellets were sieved and the pellet durability index (PDI) was determined by calculating the percentage ratio of the particles retained on a 2 mm sieve relative to the initial pellet mass [33].

(4) Aspect ratio: Aspect ratio is determined as the ratio of the diameter to the length and is an important factor for describing a cylindrical pellet's shape [76]. The strength and moisture resistance capacity are dependent on the aspect ratio of fuel pellets. The pellets with the largest diameters tend to show the poorest mechanical properties. Additionally, an increase in the diameter of the pellet could result in a lower moisture resistance capacity. The greater aspect ratio can lead to higher pellet durability, which is probably because of the improved bonding between the particles and the lower hygroscopicity of the pellets [77].

## 4. Conclusions

Carbon-neutral and renewable energy sources are the way of the future; however, biomass-derived fuels have not been widely introduced into commercial and industrial-scale applications. One commercially available biofuel is wood pellets. Alongside using woody biomass as the feedstock for pellet production, other types of biomass such as agricultural waste, MSW, and algal biomass can also be used as renewable sources for preparing fuel pellets. HTC is a hydrothermal treatment and can be used for processing different types of biomass. Because it does not require a pre-drying stage, HTC might be an ideal conversion route for biomass, particularly biomass with a high water content. During HTC, hydrochar is generated as the main product. Following this, the hydrochar can be further processed to prepare fuel pellets by a pelletizing process. After some key properties of the pellets have been characterized to ensure that the hydrochar pellets can meet the standards, they can be either used for residential heating or for co-firing power plants. In this mini-review, the effects of various parameters of the HTC reaction, including the characteristics of the biomass, the temperature, the residence time, and the feedstock: water ratio on the yield and properties of the hydrochar have been discussed. Next, the key parameters of the pelletizing process such as the temperature, applied pressure, moisture content, and binder type and dosage were described. Finally, several key specifications including the bulk density, the aspect ratio, the moisture resistance capacity, and the pellets' strength and durability for commercialization of the hydrochar pellets were covered.

**Author Contributions:** Conceptualization, R.G. and Y.H.; methodology, Y.H.; software, A.A.F.; validation, S.H.; formal analysis, S.H. and K.K.; investigation, A.A.F.; resources, Y.H.; data curation, R.G. and Y.H.; writing—original draft preparation, R.G.; writing—review and editing, Y.H., A.A.F. and S.H.; visualization, K.K.; supervision, Y.H.; project administration, Y.H.; funding acquisition, R.G. and Y.H. All authors have read and agreed to the published version of the manuscript.

**Funding:** This research was funded by Discovery Grants Program, Natural Science and Engineering Research Council of Canada (NSERC), RGPIN-2022-03203.

**Institutional Review Board Statement:** Not applicable.

**Informed Consent Statement:** Not applicable.

**Data Availability Statement:** The data presented in this study are available upon request from the corresponding author.

**Acknowledgments:** The authors would like to acknowledge funding from the NSERC through the Undergraduate Student Research Awards awarded to Rhea Gallant and the Discovery grant awarded to Yulin Hu.

**Conflicts of Interest:** The authors declare no conflict of interest.

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
