# Peer review of "A Mini-Review: Biowaste-Derived Fuel Pellet by Hydrothermal Carbonization Followed by Pelletizing"

_sustainability, doi:10.3390/su141912530_

Round 1
Reviewer 1 Report
The paper “A Mini Review: Biowaste Derived Fuel Pellet by Hydrothermal Carbonization Followed by Pelletizing” presents a literature review of the recent studies using hydrothermal carbonization as a pre-treatment method for producing hydrochar and its potential application as a solid fuel pellet. I consider that this is an interesting review with important information. However, there are several points that make the review unclear. Most of them are important point that were not deeply investigated by authors.
1) Authors should carefully proofread the paper to avoid mistakes. There are several mistakes in the paper, including references that were not linked with the text, words that are wrong and sentences that require a major attention. For instance, In section 2.1. “A general flow diagram describing of HTC of biomass is depicted in Error! Reference source not found..”. What flow diagram? What reference? In the same sentence, “The recent studies from Year 2020 to Year 2022 are summarized in”. Authors should adjust these and many other points in the manuscript.
2) Figure 1 and 2 don’t say nothing. I suggest only including figures that really can contribute for the review.
3) In table 1 the authors should include much more information, such as, residence time, severity factor, if the process is continuous, semi-continuous, or batch, the flow for non-batch process, solvent-to-feed ratio, and many others. The yield of the hydrochar should be provided.
4) What is the purpose of add the composition of the biomass? If the aim of the review is provide an overview of the biochar production, authors should associate each biomass with the yield of biochar obtained. That is, it is more interesting to see the composition after HTC.
5) After HTC, the product remained in the reactor will not be necessarily hydrochar. For example, it is clear that in Fig 5, the increase in hydrochar yield is associated with lower temperature. But at lower temperature (e.g., 180 °C) you have a lower degradation of lignocellulose. So, the product represented in the figure will not be necessarily hydrochar.
Author Response
Manuscript ID: sustainability-1934214
Title: A Mini Review: Biowaste Derived Fuel Pellet by Hydrothermal Carbonization Followed by Pelletizing
Authors: Rhea Gallant, Aitazaz A. Farooque, Sophia He, Kang Kang, Yulin Hu *
The authors are grateful to the referees for reviewing our manuscript submitted to the Special Issue “Frontier in Bio-Energy Production and Applications” at Sustainability We have revised the manuscript carefully to address all questions/comments from the referees. Detailed actions taken in the revisions are listed below. For easier tracking, “Track Changes” are used in the revised manuscript.
Reviewer-1:
The paper “A Mini Review: Biowaste Derived Fuel Pellet by Hydrothermal Carbonization Followed by Pelletizing” presents a literature review of the recent studies using hydrothermal carbonization as a pre-treatment method for producing hydrochar and its potential application as a solid fuel pellet. I consider that this is an interesting review with important information. However, there are several points that make the review unclear. Most of them are important point that were not deeply investigated by authors.
1) Authors should carefully proofread the paper to avoid mistakes. There are several mistakes in the paper, including references that were not linked with the text, words that are wrong and sentences that require a major attention. For instance, In section 2.1. “A general flow diagram describing of HTC of biomass is depicted in Error! Reference source not found..”. What flow diagram? What reference? In the same sentence, “The recent studies from Year 2020 to Year 2022 are summarized in”. Authors should adjust these and many other points in the manuscript.
Response: Thank you for this comment. The manuscript was thoroughly revised.
2) Figure 1 and 2 don’t say nothing. I suggest only including figures that really can contribute for the review.
Response: Thank you for this comment. Figure 1 and Figure 2 were removed from the revised manuscript.
3) In table 1 the authors should include much more information, such as, residence time, severity factor, if the process is continuous, semi-continuous, or batch, the flow for non-batch process, solvent-to-feed ratio, and many others. The yield of the hydrochar should be provided.
Response: As suggested by the reviewer, reactor configuration and the highest hydrochar yield were added to the Table 1 in the revised manuscript.
4) What is the purpose of add the composition of the biomass? If the aim of the review is provide an overview of the biochar production, authors should associate each biomass with the yield of biochar obtained. That is, it is more interesting to see the composition after HTC.
Response: The authors agree with the reviewer that it is important to link each biomass with the resulting hydrochar since biomass characteristics is one of important factors will affect the yield and property of hydrochar. In the revision, the influence of biomass characteristics on HTC biomass was discussed. In addition, the composition of hydrochar obtained from HTC of different types of biomass was compared with original feedstock, as summarized in Table 2.
5) After HTC, the product remained in the reactor will not be necessarily hydrochar. For example, it is clear that in Fig 5, the increase in hydrochar yield is associated with lower temperature. But at lower temperature (e.g., 180 °C) you have a lower degradation of lignocellulose. So, the product represented in the figure will not be necessarily hydrochar.
Response: The authors agree with the reviewer that lower biomass degradation is expected to have at lower temperatures, rendering a higher hydrochar yield (the solid yield). Higher temperatures make more biomass components to be degraded to small molecules. Aside from hydrochar, HTC also produces water-soluble product (water phase) and gas phase (very trace amount). While Figure 5 only shows the hydrochar yield not includes other products’ yield since this sub-section aims to explain how temperature affects the yield and properties of hydrochar rather than water and gas phases.
Reviewer 2 Report
1. As a review paper the authors should covered a little bit further on HTC. For example, catalytic effects on the process should be handled.
2. On the potential use of hydrochar, the authors seems mixed biofuel purpose with biochar purpose. As far as I understand the biochar needs an extra process such as pyrolysis. If the authors want to use the terminology "hydrochar" in both concepts, they should study and add the biochar process also.
3. On the severity factor (2.3), I'd like to recommend to cover the yield rate also in terms of severity factor.
4. I'd like to recommend also to change the title of <2.4 solid loading> to <moisture/water content>.
5. I cannot agree following statement on ash content increase since I believe that the absolute amount of the ash and the carbon do not change but the ratio of them increase since the total amount of the solid itself is decreased thru HTC process:
HTC temperature also significantly influences the property of hydrochar. Nakason et al. [47] found that the ash content of hydrochar increased with increasing temperature, which can be explained by: (i) other biomass constituents like volatiles decreased at higher temperatures and thus might leads to a higher proportion of ash in the hydrochar; and (ii) some inorganics might re-adsorb on the surface of the hydrochar.
Author Response
Manuscript ID: sustainability-1934214
Title: A Mini Review: Biowaste Derived Fuel Pellet by Hydrothermal Carbonization Followed by Pelletizing
Authors: Rhea Gallant, Aitazaz A. Farooque, Sophia He, Kang Kang, Yulin Hu *
The authors are grateful to the referees for reviewing our manuscript submitted to the Special Issue “Frontier in Bio-Energy Production and Applications” at Sustainability We have revised the manuscript carefully to address all questions/comments from the referees. Detailed actions taken in the revisions are listed below. For easier tracking, “Track Changes” are used in the revised manuscript and the revisions are highlighted in red color.
Reviewer-2:
- As a review paper the authors should covered a little bit further on HTC. For example, catalytic effects on the process should be handled.
Response: As suggested by the reviewer, additional sentences were added to the revised manuscript to cover the introduction of catalyst in the HTC process,
“2.5 Catalyst
Catalyst is often applied to facilitate biomass thermal degradation. Both homogenous and heterogenous catalysts have been used. Hydrolysis is a very important reaction involved in HTC and adding acid catalysts can promote hydrolysis. Aside from acid catalysts, alkaline catalysts like carbonate, hydroxide, and bicarbonate have demonstrated beneficial impacts on bio-oil production by deterring char formation. Thus, alkaline catalysts have been commonly applied in the bio-oil production process like HTL. Nevertheless, only few studies have applied catalyst in the HTC process for enhancing hydrochar production like Evcil et al. [61] using a mixture of AlCl3 and HCl and Wikberg et al. [62] using H2SO4”.
- On the potential use of hydrochar, the authors seems mixed biofuel purpose with biochar purpose. As far as I understand the biochar needs an extra process such as pyrolysis. If the authors want to use the terminology "hydrochar" in both concepts, they should study and add the biochar process also.
Response: Thank you for this comment. The authors agree with the reviewer that the biochar production methods are different from hydrochar. Pyrolysis or carbonization is used for preparing biochar; while hydrotreatment like HTC is used for hydrochar preparation. To avoid confusion, only hydrochar is used throughout the revised manuscript.
- On the severity factor (2.3), I'd like to recommend to cover the yield rate also in terms of severity factor.
Response: Thank you for this comment. Additional sentences were added to the revised manuscript, “In terms of the influence of severity factor on hydrochar yield, Jeder et al. [57] found that the yield of hydrochar continuously dropped with increasing severity factor from 4.0 to 7.6. This decrease in hydrochar yield is due to the carbon enrichment and the reductions in the O and H contents.”
- I'd like to recommend also to change the title of <2.4 solid loading> to <moisture/water content>.
Response: This was addressed in the revised manuscript.
- I cannot agree following statement on ash content increase since I believe that the absolute amount of the ash and the carbon do not change but the ratio of them increase since the total amount of the solid itself is decreased thru HTC process:
HTC temperature also significantly influences the property of hydrochar. Nakason et al. [47] found that the ash content of hydrochar increased with increasing temperature, which can be explained by: (i) other biomass constituents like volatiles decreased at higher temperatures and thus might leads to a higher proportion of ash in the hydrochar; and (ii) some inorganics might re-adsorb on the surface of the hydrochar.
Response: First, the weight percentage of ash and carbon in hydrochar is different from biomass since some fraction of original biomass is decomposed. Second, the absolute amount of ash (when you consider the mass loss during HTC reaction) might change after HTC reaction because some ash might retain in the solid fraction (stay in hydrochar) or some ash might go to water phase. Water phase is another product that can be produced from HTC. Or, as discussed by Nakason et al. [47], ash content or ash fraction in the biomass might increase with increasing HTC temperature since more volatile matter of original biomass can be decomposed. Another explanation for this increase in ash content could be due to the re-adsorption of ash to the hydrochar since hydrochar will have a porous structure at higher processing temperature, so some ash or inorganics might be adsorbed by the hydrochar and retain on the surface of the hydrochar. To avoid the confusion, the relevant sentences were re-written in the manuscript, “HTC temperature also significantly influences the property of hydrochar. Nakason et al. [47] found that the ash content of hydrochar increased with increasing temperature, which can be explained by: (i) other biomass constituents like volatiles are driven off as gases at higher temperatures and thus might leads to a higher proportion of ash in the hydrochar; and (ii) some inorganics might re-adsorb on the surface of the hydrochar since hydrochar would exhibit a porous structure at higher HTC temperatures. On the other hand, it should be noted that some ash or inorganic fraction of original biomass could partition into the water phase during HTC treatment.”
Round 2
Reviewer 1 Report
The paper can be accepted for publication.
Author Response
All comments/concerns given by reviewer 1 were addressed accordingly. The revised manuscript was revised again for spelling and grammar checking.
Reviewer 2 Report
Some minor change, spelling check needed. Followings are a couple of examples.
1. Pls correct below sentence at page 2.. it seems some words are missing after the word 's'.
To date, some previous literature has reviewed the use of hydrochar for improving plant growth [16], facilitating biomethane production in anaerobic digestion (AD) [17], heavy metal removal [18], gas cleanup [18], and s.
2. At Table 4
time (t,h) ---> time t (h)
Author Response
Manuscript ID: sustainability-1934214
Title: A Mini Review: Biowaste Derived Fuel Pellet by Hydrothermal Carbonization Followed by Pelletizing
Authors: Rhea Gallant, Aitazaz A. Farooque, Sophia He, Kang Kang, Yulin Hu *
The authors are grateful to the referees for reviewing our manuscript submitted to the Special Issue “Frontier in Bio-Energy Production and Applications” at Sustainability We have revised the manuscript carefully to address all questions/comments from the referees. Detailed actions taken in the revisions are listed below. For easier tracking, “Track Changes” are used in the revised manuscript and the revisions are highlighted in red color.
Reviewer-2:
Some minor change, spelling check needed. Followings are a couple of examples.
Response: Thank you for this comment. This manuscript was revised again to correct the spelling and grammatic mistakes.
- Pls correct below sentence at page 2.. it seems some words are missing after the word 's'.
To date, some previous literature has reviewed the use of hydrochar for improving plant growth [16], facilitating biomethane production in anaerobic digestion (AD) [17], heavy metal removal [18], gas cleanup [18], and s.
Response: This was addressed in the revised manuscript.
- At Table 4
time (t,h) ---> time t (h)
Response: This was addressed in the revised manuscript.